# Redefining transcriptional regulation of the *APOE* gene and its association with Alzheimer's disease

Eun-Gyung Lee[1], Jessica Tulloch[1], Sunny Chen[1], Lesley Leong[1], Aleen D. Saxton[1], Brian Kraemer[1,2], Martin Darvas[3], C. Dirk Keene[3], Andrew Shutes-David[1], Kaitlin Todd[1], Steve Millard[1], Chang-En Yu[1,2]*

1 Geriatric Research, Education, and Clinical Center, VA Puget Sound Health Care System, Seattle, WA, United States of America, 2 Department of Medicine, University of Washington, Seattle, WA, United States of America, 3 Department of Pathology, University of Washington, Seattle, WA, United States of America

* changeyu@uw.edu

**Data Availability Statement:** All relevant data are within the manuscript and its Supporting Information files.

**Funding:** This work was supported in part by the U. S. Department of Veterans Affairs Office of

## Abstract

The apolipoprotein E gene (*APOE*) is the strongest genetic risk factor for late-onset Alzheimer's disease (AD), yet the expression of *APOE* is not clearly understood. For example, it is unclear whether AD patients have elevated or decreased *APOE* expression or why the correlation levels of *APOE* RNA and the ApoE protein differ across studies. Likewise, *APOE* has a single CpG island (CGI) that overlaps with its 3'-exon, and this CGI's effect is unknown. We previously reported that the *APOE* CGI is highly methylated in human post-mortem brain (PMB) and that this methylation is altered in AD frontal lobe. In this study, we comprehensively characterized *APOE* RNA transcripts and correlated levels of RNA expression with DNA methylation levels across the *APOE* CGI. We discovered the presence of *APOE* circular RNA (circRNA) and found that circRNA and full-length mRNA each constitute approximately one third of the total *APOE* RNA, with truncated mRNAs likely constituting some of the missing fraction. All *APOE* RNA species demonstrated significantly higher expression in AD frontal lobe than in control frontal lobe. Furthermore, we observed a negative correlation between the levels of total *APOE* RNA and DNA methylation at the *APOE* CGI in the frontal lobe. When stratified by disease status, this correlation was strengthened in controls but not in AD. Our findings suggest a possible modified mechanism of gene action for *APOE* in AD that involves not only the protein isoforms but also an epigenetically regulated transcriptional program driven by DNA methylation in the *APOE* CGI.

## Introduction

The ε4 variant of the human *APOE* gene is a well-established genetic risk factor for the late-onset AD. This gene's protein product, ApoE, plays a key role in lipid metabolism. Human ApoE is a polymorphic protein, and the presence of either arginine or cysteine at amino acid positions 112 and 158 defines its three common protein isoforms: E2, E3, and E4. These iso-forms vary in their affinity for lipoprotein particles and low-density lipoprotein receptors [1],

Research and Development Biomedical Laboratory Research Program and National Institute of Health/National Institute on Aging grants (1) Grant number: I01BX000933 (VA Merit Review, PI: Chang-En Yu, PhD) and (2) Grant number: 1R01AG062514-01 (NIH/NIA, PI: Martin Darvas, PhD). The funders had no role in study design, data collection and analysis, decision to publish, or preparation of the manuscript

**Competing interests:** The authors have declared that no competing interests exist.

leading to isoform-specific differences in total serum cholesterol levels [2]. The ApoE E2/E3/E4 isoforms are encoded by the ε2, ε3, and ε4 genetic variants, respectively, which are driven by two single-nucleotide polymorphisms (SNPs, rs429358 and rs7412) in the *APOE* coding region [3, 4]. These protein variations, and their corresponding ε2/ε3/ε4 genetic variants, have been linked with differential risks of cardiovascular diseases [5, 6] and late-onset AD [7–10]. Inheritance of the ε4 allele increases the risk of developing AD in a gene dose–dependent manner and predisposes carriers to an earlier age of onset [7, 11]. Persons with one copy of the ε4 allele have a three times greater risk of developing AD than persons with two copies of the ε3 allele, and persons with two copies of ε4 allele have an 8- to 12-fold increase in risk [12] https://alz.org/media/Documents/alzheimers-facts-and-figures-2019-r.pdf. In opposition, inheritance of the ε2 allele appears to have a modest protective effect against developing AD [12, 13]. Although genome-wide approaches (e.g., the use of SNP arrays or massive parallel sequencing) have identified more than 25 additional loci that are associated with AD, these loci have only moderate effect sizes with odds ratios ranging from 1.1 to 1.5, whereas the odds ratio for *APOE* ε4 versus ε3 is estimated at 3.7 [14]. Thus, the ε4 allele of *APOE* remains the most common and influential genetic risk factor for developing AD [15], suggesting that variation in *APOE* manifests strong biological effects in the etiology of AD.

Biomedical researchers have traditionally followed the protein-centric view, and thus, mainstream research on *APOE* has focused on protein isoform–specific differences in structure and function. This work on ApoE's protein isoform has generated valuable insights into how ApoE4 protein might increase the risk of AD. Various hypotheses have been proposed, including Aβ aggregation and clearance [16], protein domain interaction and neurotoxicity [17], neuroinflammation [18], tau hyperphosphorylation [19], and differential lipidation states [20]. But the main function of the ApoE protein, the redistribution of lipoproteins and cholesterol, cannot readily explain the complex pathophysiology of AD, and the precise molecular mechanisms by which ApoE4 exerts its detrimental effect in AD remain elusive. Still, meta-analyses that encompass many studies and ethnic groups have robustly identified that *APOE* harbors an overwhelming effect in AD risk; therefore, additional effects of *APOE*, beyond its role in coding for the ApoE protein, likely contribute to AD risk as well.

Altered *APOE* expression, both in mRNA and protein, might explain some of *APOE*'s effect in AD. In the periphery, the liver produces the majority of circulating ApoE, which binds lipids and interacts with receptors to initiate cellular uptake of lipoprotein particles [21]. In the central nervous system (CNS), glial cells produce the majority of ApoE [22], which promotes the transport of lipids to neurons and thereby conducts important functions in neuronal maintenance, repair, and brain homeostasis [23]. Brains from persons with AD show aberrant lipid metabolism [24], which suggests that protein isoform–dependent differences in steady-state ApoE levels could influence *APOE*-related AD risk. Indeed, the ε4 allele encodes less ApoE than its ε2/ε3 counterparts, and it likewise provides insufficient levels of functional ApoE4 to maintain synaptic remodeling and cholesterol homeostasis in the CNS [25–28]. In contrast, ε2 carriers have higher ApoE levels than both ε3 and ε4 carriers [28], and these higher ApoE levels may help protect against the late-life development of AD. However, studies on ApoE levels in AD have always yielded conflicting results. For example, low plasma ApoE levels are consistently linked to an increased risk of AD and dementia [29, 30], whereas ApoE levels in the AD CNS have been described as up-regulated [31–33], down-regulated [34–36], or unchanged [37, 38]. Alternatively, researchers have investigated the expression levels of *APOE* RNA, and such studies have more consistently shown elevated *APOE* RNA levels in AD PMB [32, 39–42], though it is unclear whether these elevations in *APOE* RNA are ε4-specific [41, 43–45]. In addition, these studies have always identified a common challenge: *APOE* RNA expression levels are not well correlated with protein levels. That inconsistency complicates the interpretation of

*APOE*'s quantitative effects in AD and has led to speculation that *APOE* RNA's transcriptional/post-transcriptional modifications may contribute to these diverse observations across studies.

Transcriptional regulation of *APOE* is likely to occur differently than in most other genes because *APOE* carries a well-defined 5'-C-phosphate-G-3' (CpG) island (CGI) that overlaps with its last exon. The two ε2/ε3/ε4-defining SNPs are located in this region, and they also alter the CpG content by either contributing (in ε4) or disrupting (in ε2) a single CpG site, thereby affecting the DNA methylation landscape of the CGI [46]. In our prior work, we have shown that the *APOE* CGI is highly methylated in PMB and that its methylation level is decreased in AD frontal lobe [47]; this suggests that *APOE*'s effects in AD also involve CGI methylation–modulated transcription. In our current work, we have discovered and developed assays for two new species of *APOE* transcripts: circular RNA (circRNA) and full-length mRNA. These new transcripts are inherently included in current assay measures of *APOE* total mRNA and have not been distinguished previously. The aim of this study is to assess the relationship between various RNA transcripts in the PMB of AD and control subjects, to define the *APOE* CGI's role in *APOE* RNA transcription, and to characterize how the interaction between RNA expression and DNA methylation correlates with the risk of developing AD.

## Materials and methods

### Human PMB and cell lines

This work does not involve any human subject recruitment or intervention; it uses deidentified human biospecimen that have been collected by other established programs; therefore, no consent was obtained for this work. All human specimens were obtained from the University of Washington (UW) Alzheimer's Disease Research Center after approval by the institutional review board of the Veterans Affairs Puget Sound Health Care System (MIRB# 00331). AD patient diagnosis was confirmed postmortem by neuropathological analysis. Clinically normal subjects were volunteers who were ≥65 years of age, were never diagnosed with AD, and lacked AD neuropathology at autopsy. Brains from AD volunteers exhibited Braak stages between V and VI, whereas brains from control subjects exhibited Braak stages between 0 and III. Postmortem frontal lobe from the middle frontal gyrus and cerebellar tissues were obtained from brain tissue that had been rapidly frozen at autopsy (<10 hours after death) and stored at –80˚C until use. Hepatocytoma HepG2 and glioblastoma U87 cells (ATCC) were grown in 89% Dulbecco's modified Eagle's medium (DMEM) (Gibco); neuroblastoma SH-SY5Y cells (ATCC) were grown in 89% DMEM with F12. Both media were supplemented with 10% fetal bovine serum (Gibco). Glioblastoma LN-229 cells (ATCC) were grown in 94% DMEM supplemented with 5% FBS. All cell cultures were supplemented with 1% penicillin/streptomycin and cultured at 37˚C in a 5% $CO_2$ atmosphere.

### DNA/RNA extraction and *APOE* genotyping

Genomic DNA was isolated from frozen PMB using the AllPrep DNA/RNA Mini Kit (Qiagen). Nucleic acid concentrations were measured by NanoPhotometer (Implen), and samples were stored at –20˚C prior to use. The ε2/ε3/ε4 alleles of *APOE* were genotyped using two TaqMan allele discrimination assays (C_3084793_20 and C_904973_10, Thermo Fisher Scientific). All procedures were performed according to the manufacturers' protocols.

### Cellular Fractionation: Preparation of cytoplasmic and nuclear RNA

Cultured cells were washed and harvested in phosphate-buffered saline (PBS) with a cell scraper. Following centrifugation, cold Cell Disruption Buffer (10 mM KCl, 1.5 mM $MgCl_2$, 20

mM Tris-HCl, 1 mM DTT) was added to the cell pellet and incubated on ice for 10 minutes prior to homogenization using a glass homogenizer. Cell lysis was confirmed using a phase-contrast microscope. Triton x-100 was added to the homogenate at a final concentration of 0.1% prior to centrifugation at 1500 g for 5 minutes. The supernatant, containing the cytoplasmic portion, was transferred and processed using a RNeasy-mini RNA isolation kit (Qiagen). The remaining nuclear pellet was processed using TRIzol RNA isolation.

## Western blot and enzyme-linked immunosorbent assay (ELISA)

Semiquantitative Western blot analysis was performed on cell lysates from HepG2 cells using antibodies specific to either the N-terminus (ab51015, Abcam) or the C-terminus (ab52607, Abcam) of ApoE. β Actin (ACTB) antibody was used as a loading control. Relative quantification of the HepG2 cell lysate was measured by densitometry using a standard curve method. We quantified ApoE levels in culture medium using an enzyme-linked immunosorbent assay (ELISA) according to the manufacturer's instructions (Mabtech, #3712-1HP-2). Media was collected from three independent 5-aza-Dc treated and three untreated HepG2, LN229, and SHSY5Y cell cultures (18 total). Culture medium was refreshed at 48 hrs and collected at 72 hrs. Media was cleared by centrifuging at 1500 rpm for 10 min at 4˚C, supernatant was collected and frozen in two aliquots. Due to the high level of ApoE expression in HepG2 cells, media was diluted 1:200 in Apo ELISA buffer prior to assaying. Media from LN229 and SHSY5Y cultures was not diluted. All samples were run in duplicate and concentrations were averaged to account for technical variation. ApoE levels in the media were normalized by total cell number to account for cell death caused by 5-aza-dC treatment.

## Reverse transcriptase (RT) reaction and quantitative PCR (qPCR) assay

Total RNA (100 ng) was used for each 20 μL RT reaction, and cDNA synthesis was performed using the PrimeScript RT Reagent Kit (Takara Bio USA). The resulting cDNA was diluted 20 times for qPCR, and expression levels were measured by qPCR using TaqMan assays in an ABI Prism 7900 Sequence detector (Applied Biosystems). Each 10 μl reaction contained a fixed RNA input (5 ng), 0.5 μl of the 20x TaqMan assay, and 5 μl of 2x TaqMan Universal PCR Reaction Mix (Thermo Fisher). The thermal cycling program consisted of 2 minutes at 50˚C, 10 minutes at 95˚C, and then 40 cycles of 15 seconds at 95˚C and 1 minute at 60˚C. The amplification efficiency of *APOE* TaqMan assays, including for circular, full-length, and total RNA, was measured by a standard curve method using each serial dilution in qPCR reactions. The calculated amplification efficiency using $[10^{(-1/\text{slope})}]$ -1 is 0.95 (circRNA), 0.90 (full-length RNA), and 0.91 (total RNA), respectively. Information on primers, probes, and TaqMan assays is listed in S1 Table. For each sample, two independent RT reactions were performed for cDNA prep, and triplicate in qPCR assay were performed for each cDNA prep. For *APOE* RNA quantification, all reactions were quantified by using a fixed threshold (0.15) in the linear range of amplification and recording the number of cycles (cycle threshold, Ct) required for the fluorescence signal to cross the threshold. To control for the quantity of input RNA, we quantified *ACTB* mRNA as an internal control for each sample by obtaining a normalized ΔCt value: Ct (target)–mean of the *ACTB* Ct triplicate. In this setting, smaller ΔCt values indicate higher RNA expression levels. Additionally, fold change (FC) between expression levels for total RNA versus circRNA or full-length mRNA was computed as FC (target) = $2^{-\Delta\Delta Ct}$, where ΔΔCt = ΔCt (target)– ΔCt (total) [48]. For truncated mRNA, FC was computed as FC (truncated) = 1 –FC (full-length)–FC (circular).

## Genomic walking by qRT-PCR

This analysis was carried out by the following procedures. RT reactions were performed using a set of evenly spaced primers to prime either the sense or antisense transcript of *APOE* exon 4. These cDNAs were subsequently amplified by PCR reactions using a series of spaced upstream primers that were designed to pair with each RT primer and thereby amplify their corresponding cDNAs. All RT-PCR reactions were repeated three times. All cDNA products were fully characterized using gel purification and DNA sequencing.

## 5-aza-dC treatment

Twenty-four hours prior to treatment, the cells were seeded at a density of 30%. They were then treated with 250 nM 5-aza-dC, which was replenished every 24 hours with fresh media. For controls, the same number of cells were plated and cultured without 5-aza-dC. Cell pellets were collected 72 hours post-treatment and subjected to genomic DNA and total RNA isolation, followed by measurement of DNA methylation and RNA expression. The cell supernatants were collected after centrifugation of cell culture media at 1500 rpm, 10 min, 4°C. The cell supernatants were used for quantification of ApoE protein secreted from cells by ELISA.

## Bisulfite pyrosequencing

Quantification of DNA methylation levels by pyrosequencing was performed. Briefly, genomic DNA (500 ng each) was bisulfite converted using the EpiTect Bisulfite Kit (Qiagen). To evaluate the methylation status of the *APOE*'s DMR 1, we designed pyrosequencing assays to cover 27 CpG sites (CpG 11–37). PCR was performed on approximately 200 ng of bisulfite-converted DNA using PyroMark PCR kits (Qiagen) on a GeneAmp PCR System 9700 (Applied Biosystems, Grand Island, NY). Pyrosequencing was carried out on a PyroMark Q24 system (Qiagen) and data was analyzed using PyroMark Q24 software, version 2.0.6 (Qiagen). Bisulfite treatment controls were integrated as a quality control measure. Detailed procedures have been described in our prior published works [47]. Biological samples untreated and two independently treated with 5-aza-dC were used for bisulfite pyrosequencing.

## Chromatin immunoprecipitation quantitative PCR (ChIP-qPCR) assay

We trypsinized the HepG2, LN-229 and SH-SY5Y cells ($\sim$ 3-5x10$^6$ each) and cross-linked proteins to DNA by adding 1% formaldehyde in culture media. After cell lysis, we applied sonication to shear genomic DNA into 400-to-700-bp fragments using a Bioruptor sonicator (Diagenode), followed by centrifugation to remove cell debris. For immunoprecipitation, we incubated 1 μg of chromatin with 0.8 μg of rabbit polyclonal anti-MECP2 antibody (ab2828, Abcam) overnight. We then purified protein-enriched DNA fragments using a Magnetic One-step Kit (Abcam) and performed qPCR using SYBR Green on an ABI 7900 Sequence detector. We tested *APOE* gene regions using absolute quantification with a standard curve method to calculate the copy numbers of the recovered DNA fragments. All ChIP-qPCR assays were performed 3 times using cells independently treated with 5-aza-dC.

## Statistical analysis

RNA expression was measured using mean ΔCt for each subject/tissue type/RNA type (circular, full-length, total) combination. For each tissue type, Pearson product-moment correlations were used to assess the association of expression levels between RNA types. To assess the relationship between RNA expression and RNA type (i.e., RNA species) in PMB of AD and control subjects, we first tested whether differences in expression between AD and control PBM in

each RNA type varied by tissue type using a linear model with ΔCt as the response (dependent) variable and terms for disease status, tissue type, and a disease status–by–tissue type interaction. Next, for each RNA type and tissue type, we tested for a difference in RNA expression between AD and control PMB using a linear model that included a term for disease status. We used the Holm correction for multiple comparisons [49] to adjust for six comparisons (two tissue types x three RNA types). We calculated the effect size as (Difference / Estimated Population Standard Deviation based on residuals), and we computed the confidence interval for effect size based on equation (16) of Nakagawa and Cuthill [50]. We also looked at models that included the covariates sex and *APOE* ε4 status (ε4+ vs. ε4−). To assess the relationship between *APOE* total RNA expression and mean DNA methylation levels across the *APOE* CGI's AD DMR1 (CpG sites #11–37) [46], for each tissue type we looked at (1) a linear model that included total RNA ΔCt as the response (dependent) variable and a term for mean DNA methylation level across the DMR 1, using both AD and control subjects combined, (2) a model that also included a term for disease status and a disease status–by–mean DNA methylation interaction term, and (3) separate models for AD and control subjects. We also created the same models using DNA methylation at each of the 27 DMR 1 CpG sites. For these latter models, we used the Holm adjustment for multiple comparisons. All statistical analyses were carried out using R [51] and the packages *GGally* [52], *tidyverse* [53], and *EnvStats* [54].

## Results

### Presence of multiple mRNA transcripts in *APOE*

The gene structure of *APOE* consists of four exons, with an open reading frame (ORF) extending from exons 2 to 4 and a single CGI that overlaps with exon 4. The two ε2/ε3/ε4-determining SNPs (rs429358 and rs7412) also reside in this exon 4/CGI region (Fig 1A). To fully characterize *APOE*'s mRNA, we applied an *in-silico* approach that used the Ensembl RNA database [55] to search for all documented forms of *APOE* mRNA. We found that at least four ORF-containing mRNA transcripts are present, which vary by either the promoter usage or the termination sites (Fig 1B). Among these mRNAs, only one (ENST 252486) contains the entire ORF and can produce the 299 amino acids of the full mature ApoE protein (Fig 1C).

The other mRNAs all terminate prematurely in the midsection of exon 4 with expected C-terminal endings at amino acids 216, 219, and 243. The resulting truncated proteins, if made, would not carry ApoE protein's lipid-binding domain, which is located from amino acids 244 to 272. Notably, none of these truncated mRNAs have been described in the literature; whether they are true transcripts or merely artifacts remain to be validated. Because the commercially available TaqMan assay for *APOE* mRNA (Hs00171168_m1, Thermo Fisher) targets the splice junction between exons 3 and 4, a structure and sequence that is included within both full-length and truncated mRNAs, it cannot distinguish between these two mRNA species but rather quantifies them together. Therefore, to distinguish full-length mRNA from the *APOE* RNA pool, we designed a new TaqMan assay (S1 Table) that targets the 3'UTR of *APOE* and thereby allows specific quantification of only the full-length *APOE* mRNA. Using these two TaqMan assays, we can deduce the levels of truncated mRNA by subtracting full-length *APOE* RNA from the total *APOE* RNA in the same samples.

We applied these TaqMan assays and quantified *APOE* RNA from human cell lines. Human liver cells and glial cells of the CNS are the two major cell types that express *APOE*. Thus, we selected our cell models to best mimic these cell-types including hepatocyte (HepG2) and glia (U87 and LN-229) cells. We also included a neuronal related cell (SH-SY5Y) for comparison purposes. Full-length mRNA accounted for approximately 79%, 33%, and 41% of total RNA, respectively, in these three cell lines (Fig 2A). This result suggests that full-length mRNA

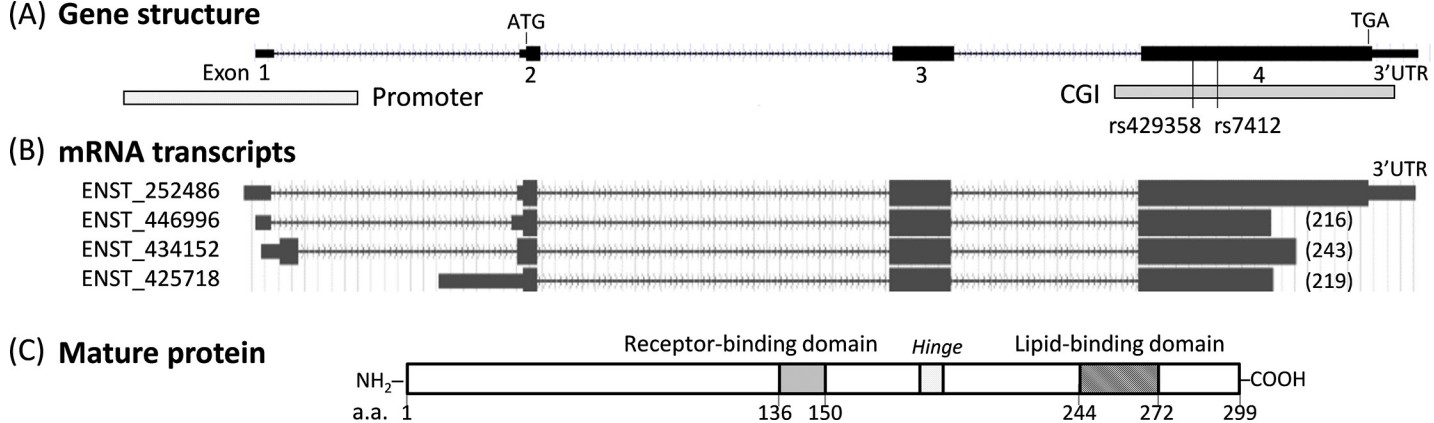

**Fig 1. Structures of the *APOE* gene, RNA transcripts, and protein.** (A) Map of *APOE* gene shows four exons, start and stop codons, two ε2/ε3/ε4-defining SNPs, promoter, and CGI. (B) Map of various mRNA transcripts defined by Ensembl database. Numbers in parentheses indicate the amino acid position at the mRNA's 3' ending points. (C) Map of mature ApoE protein with domain structures. Numbers indicate amino acid positions. CGI: CpG island; SNP: single-nucleotide polymorphism.

constitutes only a portion of the total RNA, with truncated mRNAs likely constituting some of the missing fraction.

Given that any protein translated from the truncated mRNA would lack a portion of its C-terminal domain, we used antibodies specific to either the N-terminus or the C-terminus of the ApoE protein and performed semiquantitative Western blot analysis on lysates from HepG2 cells to inspect whether the truncated mRNAs can be translated into proteins. The ApoE2, E3, E4 isoforms related amino acid alterations occur at the protein's N-terminus domain and do not alter the C-terminus domain. Therefore, the C-terminus antibody can target all three isoforms of ApoE protein with equal efficiency. Both antibodies revealed similar amounts of protein at about 36 kDa, and no other protein bands were observed on the blot (Fig 2B and 2C). This result shows no evidence of a truncated ApoE protein and thus suggests that only the full-length *APOE* mRNA can be translated into functional ApoE protein. We have also run Western blots on cell lysates from SH-SY5Y and U87. However, the ApoE expression in SH-SY5Y and U87 cells was too low (estimated to be less than one percent of HepG2) to quantify using our standard curve method.

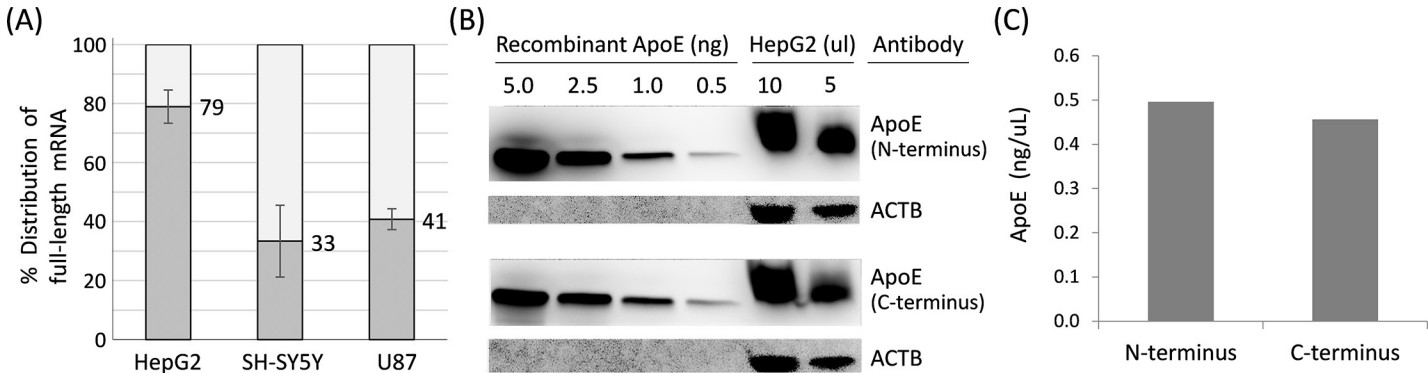

**Fig 2. qRT-PCR of the *APOE* full-length mRNA and Western blot of ApoE protein.** (A) Compositions of full-length mRNA are 79%, 33%, and 41% of total RNA in HepG2, SH-SY5Y, and U87 cells, respectively. (B) Western blots of recombinant ApoE protein and HepG2 cell culture lysates, blotted with antibodies against the N-terminus (upper panel) and C-terminus (lower panel) of ApoE or with ACTB. (C) Semiquantitative analysis on blots of 5 μl HepG2 cell culture lysates using both N- and C-terminus antibodies. ACTB: β Actin; qRT-PCR: quantitative reverse transcriptase polymerase chain reaction.

## Presence of circular RNA (circRNA) transcripts in *APOE*

In addition to the *in-silico* approach, we also conducted an RNA screening approach to search for novel *APOE* RNA transcripts that deviate from the known *APOE* mRNAs. Using a genomic walking strategy in conjunction with reverse transcriptase (RT) polymerase chain reaction (PCR), we identified two novel RNAs that differ in size, but both have an unusual structure that can only be explained by a circular formation. Both RNAs are sense-oriented with a regular splice junction between exons 3 and 4, as well as with back-splice junctions connecting a midsection of exon 4 back to the midsections of exon 3. To validate that these RNAs were indeed circRNAs and not PCR artifacts, we designed two outward-facing primers (Fig 3A and S1 Table) that were only capable of PCR amplifying circular—and not linear—templates. Using random primed RT reactions, this primer set consistently amplified these two circRNAs in total RNA that was isolated from human cell lines (HepG2, U87, and SH-SY5Y) and PMB samples (Fig 3B). A detailed analysis showed that both circRNAs carry an identical splice junction spanning exons 3 and 4 followed by 251 nucleotides (nt) of exon 4. The 3' end of this 251-nt section is then back-spliced into one of two sites on exon 3: the larger transcript (defined as L-circRNA) carries 139 nt from the 3' end of exon 3, and the smaller transcript (defined as S-circRNA), which we observed at a ~30-fold greater rate than the L-circRNA, carries 28 nt from the 3' end of exon 3 (Fig 3A). Thus, the presence or absence of an extra 111 nt on exon 3 differentiates these two circRNAs. Notably, the ε4-determining SNP (rs429358) is present in both circRNAs and can produce transcript variants (ε4 vs ε2/ε3) of each circRNA. Sequence information for these circRNAs is shown in S1A Fig. These circRNAs' back-spliced sites share a common six-nt motif (AGCTGC) that has the potential to form base pairings and stem-loop structures (S1B and S1C Fig), which may be involved in the biogenesis of these circRNAs. We developed a new TaqMan assay (S1 Table) for the circRNAs and determined their cellular distribution in HepG2 cells. Like other cellular mRNAs, the *APOE* circRNAs are mainly present in the cytoplasm with an expression level there that is about 6-fold greater than in the nucleus (Fig 3C).

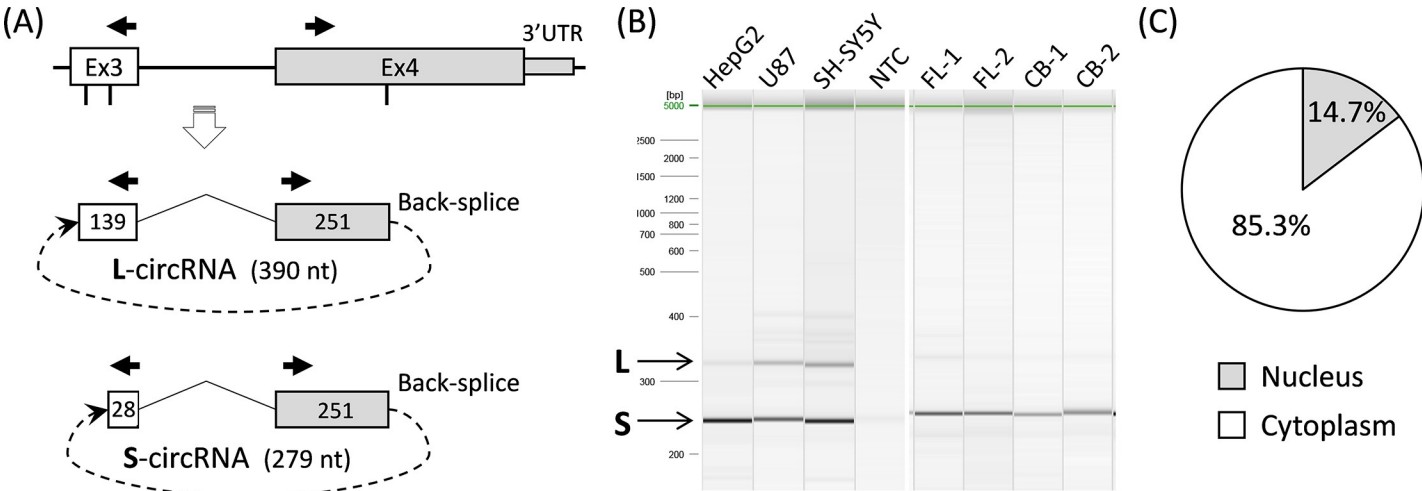

**Fig 3. Map and features of *APOE* circRNA.** (A) Diagram shows the origin of circRNA from Exs 3 and 4 of the *APOE* gene. Circular structure and size of both L- and S-circRNAs are shown. Vertical lines under exons indicate back-splice junction sites, and dotted lines indicate a back-splicing of Ex4 to Ex3. Solid arrows represent outward-facing primers used to amplify circRNAs. (B) Gel picture shows PCR products of L- and S-circRNAs from three cell lines (HepG2, U87, and SH-SY5Y) and PMB samples (FL-1, FL-2, CB-1, and CB-2]). (C) Pie chart shows that *APOE* circRNAs are present mainly in the cytoplasm (>85%) as determined by qRT-PCR on cellular fractions from HepG2 cultures. CB: cerebellum; circRNA: circular RNA; Ex: exon; FL: frontal lobe; L: large; NTC: no template control; PCR: polymerase chain reaction; PMB: postmortem brain; qRT: quantitative reverse transcriptase; S: small.

Given that these circRNAs carry the same splicing junction at exons 3 and 4 as all other *APOE* mRNA transcripts, we suspected that the commercial *APOE* TaqMan assay, mentioned above, might also be amplifying these circRNAs. We therefore generated a linear template of S-circRNA containing 279 nt (i.e., 28 nt of exon 3 plus 251 nt of exon 4, Fig 3A) and tested it in PCR reaction. We were able to confirm that this commonly used TaqMan assay *does* amplify the linear version of *APOE* circRNA and that this circRNA likely constitutes a portion of the observed discrepancy between *APOE* total RNA and full-length mRNA. Thus, we concluded that this commercial TaqMan gene expression assay of *APOE* is suitable for detecting total RNA of *APOE*, including circRNA, full-length mRNA, and truncated mRNA. Overall, the presence of multiple *APOE* RNA species suggests that transcriptional regulation of *APOE* is much more complex than previously thought.

## Expression of *APOE* RNAs in PMB

Because *APOE* is the strongest known genetic risk factor for late-onset AD, we next investigated whether various *APOE* RNAs have AD-specific expression levels in PMB tissues. The demographics of the 54 AD and 25 control subjects used for this study are listed in Table 1. We selected frontal lobe and cerebellum for this experiment because the frontal lobe is heavily affected by AD pathology whereas the cerebellum is minimally affected. We independently quantified the expression of three *APOE* RNAs (circular, full-length, and total) from AD and

**Table 1. Demographics of the study sample.**

|  | Frontal Lobe | | Cerebellum | |
| --- | --- | --- | --- | --- |
|  | AD | Control | AD | Control |
| **Participants, n** | 44 | 21 | 51 | 25 |
| **Sex: female, n (%)** | 24 (54.5) | 12 (57.1) | 28 (54.9) | 15 (60.0) |
| **APOE ε4 allele: positive, n (%)** | 31 (70.5) | 4 (19.0) | 32 (62.7) | 6 (24.0) |
| **Mean age at death: years (SD)** | 86.8 (6.9) | 87.9 (8.6) | 87.6 (6.5) | 86.6 (8.9) |
| **Mean age at AD onset: years (SD)** | 77.9 (8.1) |  | 79.1 (7.8) |  |
| **Mean disease duration: years (SD)** | 8.9 (4.4) |  | 8.4 (4.4) |  |
| **Mean postmortem interval: hours (SD)[a]** | 5.6 (2.6) | 4.8 (2.2) | 5.4 (2.5) | 4.8 (2.2) |
| **CERAD plaques, n** |  |  |  |  |
| **Absent** | 0 | 8 | 0 | 9 |
| **Sparse** | 0 | 8 | 0 | 9 |
| **Moderate** | 6 | 3 | 7 | 5 |
| **Frequent** | 38 | 2 | 44 | 2 |
| **Braak tau, n[b]** |  |  |  |  |
| **0** | 0 | 0 | 0 | 1 |
| **I** | 0 | 6 | 0 | 6 |
| **II** | 0 | 10 | 0 | 12 |
| **III** | 0 | 5 | 0 | 5 |
| **IV** | 0 | 0 | 0 | 0 |
| **V** | 15 | 0 | 20 | 0 |
| **VI** | 29 | 0 | 31 | 0 |

*Notes*: A total of 54 AD and 25 control PMB samples were used. There were 41 AD and 21 control subjects who had data for both cerebellum and frontal lobe.

Alzheimer's disease; CERAD: Consortium to Establish a Registry for Alzheimer's Disease; PMB: postmortem brain; SD: standard deviation.

[a] 1 AD subject had a missing value for the frontal lobe.

[b] 1 control subject had a missing value for the cerebellum.

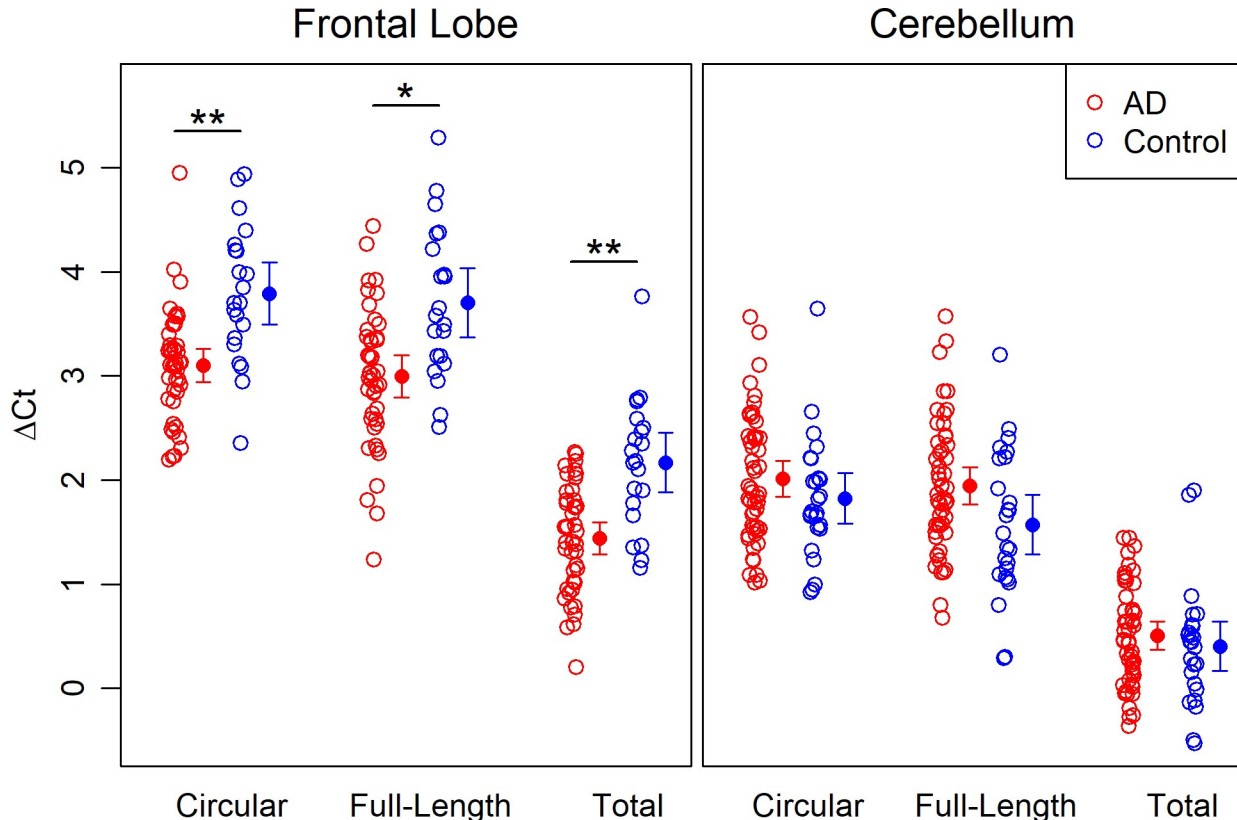

**Fig 4. *APOE* RNA expression levels in human PMB tissues.** Expression levels of three *APOE* RNAs (circular, full-length, and total) are plotted as values of ΔCt (Ct of *APOE* RNA–Ct of *ACTB* RNA) and compared between AD (red) and control (blue) in both frontal lobe (left panel) and cerebellum (right panel). Filled circles with error bars represent mean with 95% CI. *: Holm-corrected p<0.001; **: Holm-corrected p<0.0001. ACTB: β Actin; AD: Alzheimer's disease; CI: confidence interval; Ct: cycle threshold; PMB: postmortem brain.

control PMB using three TaqMan assays. The *APOE* circRNAs were present in all human PMB samples tested.

Overall, expression levels between the three *APOE* RNAs, as measured by change in threshold (ΔCt), were highly correlated (r ~0.8 to 0.9 for frontal lobe, and r ~0.9 for cerebellum, S2 Fig). For each RNA, the difference in expression between AD and control subjects varied by tissue type (p<0.0001 for all three interactions). Total RNA had the highest expression levels (i.e., lowest ΔCt) among the RNAs in both tissues, and all three RNAs had lower expression levels in the frontal lobe when compared with the cerebellum (Fig 4). Statistical models that considered tissue type separately showed significantly higher expression in AD (all Holm-corrected p<0.001) compared to controls in the frontal lobe, whereas there was no difference in expression between AD and controls in the cerebellum (Fig 4 and Table 2). In the frontal lobe, differences in expression between AD and control subjects, and the associated effect sizes, were similar for all three RNAs (effect sizes ranging from 1.04 to 1.32, Table 2), suggesting that their individual expression levels have a similar effect in AD.

When *APOE* ε4 status and sex were included as covariates, effect sizes increased slightly for all three RNAs, though the increases were more pronounced for circRNA (from 1.21 to 1.45) and total RNA (from 1.32 to 1.43) than for full-length mRNA (from 1.04 to 1.05). Although this result did not reach statistical significance, it could indicate that expression of circRNA may be further modulated by *APOE* genotype and/or gender.

**Table 2. Expression levels (ΔCt) of *APOE* RNA types in PMB.**

| Tissue | Frontal Lobe | | | Cerebellum | | |
|---|---|---|---|---|---|---|
| RNA | Circular | Full-length | Total | Circular | Full-length | Total |
| **Mean (SD)** | | | | | | |
| **AD** | 3.10 (0.53) | 2.99 (0.67) | 1.44 (0.51) | 2.01 (0.61) | 1.95 (0.64) | 0.51 (0.48) |
| **Control** | 3.79 (0.65) | 3.71 (0.73) | 2.17 (0.63) | 1.83 (0.59) | 1.57 (0.70) | 0.41 (0.57) |
| **Without Covariates** | | | | | | |
| **Difference (SE)** | 0.69 (0.15) | 0.71 (0.18) | 0.73 (0.15) | -0.19 (0.15) | -0.37 (0.16) | -0.10 (0.13) |
| **p-Value (Holm)** | <0.0001 (0.0001) | 0.0002 (0.0009) | <0.0001(<0.0001) | 0.21 (0.43) | 0.02 (0.07) | 0.42 (0.43) |
| **95% CI** | [0.39, 0.99] | [0.35, 1.08] | [0.44, 1.02] | | | |
| **Effect Size [95% CI][a]** | 1.21 [0.67, 1.75] | 1.04 [0.51, 1.56] | 1.32 [0.78, 1.87] | | | |
| **With Covariates[b]** | | | | | | |
| **Difference (SE)** | 0.83 (0.17) | 0.73 (0.21) | 0.79 (0.17) | -0.11 (0.16) | -0.29 (0.17) | -0.03 (0.13) |
| **p-Value (Holm)** | <0.0001 (<0.0001) | <0.001 (0.004) | <0.0001 (<0.0001) | 0.5 (1) | 0.09 (0.27) | 0.85 (1) |
| **95% CI** | [0.48, 1.17] | [0.31, 1.15] | [0.46, 1.13] | | | |
| **Effect Size [95% CI][a]** | 1.45 [0.90, 2.01] | 1.05 [0.52, 1.58] | 1.43 [0.88, 1.99] | | | |

Holm: p-value adjusted for multiple comparisons based on the method of Holm (6 models: two tissue types by three RNA types). AD: Alzheimer's disease; CI: confidence interval; SD: standard deviation; SE: standard error.

[a] Effect size computed as (Difference / Estimated Population SD based on residuals). CI for effect size based on equation (16) of Nakagawa and Cuthill [50].

[b] Linear model includes the covariates sex and *APOE* ε4 status.

In addition, we also estimated the levels of truncated mRNA by subtracting the levels of both circRNA and full-length mRNA from total RNA. The result showed that each of the three *APOE* RNA species (circular, full-length, and truncated) constitutes approximately one third of the total RNA in the frontal lobe; this distribution is consistent between AD and control subjects (S2 Table). The distribution pattern was slightly different in the cerebellum, which had a higher proportion of full-length and a lower proportion of truncated mRNAs, especially in the control subjects.

## Correlation between RNA expression and DNA methylation

To explore the transcriptional regulation behind *APOE* that generates multiple RNA species, we directed our attention to the *APOE* CGI that is hypermethylated in human PMB [47]. The truncated mRNAs all terminate in the midsection of this CGI; therefore, we speculated that DNA methylation of this CGI could modulate RNA production in *APOE*. To test this hypothesis, we used linear models to evaluate the relationship between the levels of *APOE* total RNA expression and DNA methylation at this CGI in PMB tissues. When looking at all samples combined (including both AD and control subjects), we observed a negative correlation between total RNA expression and mean differentially methylated region 1 (DMR 1) methylation in the frontal lobe (r = -0.28, 95% CI [-0.49, -0.04], p = 0.022, Fig 5A). This DMR 1 has been demonstrated to carry AD-specific differential methylation in the frontal lobe in our prior work [47]. When stratified by disease status, this correlation was significant in controls (r = -0.46, 95% CI [-0.74, -0.03], p = 0.037) but not in AD subjects (r = -0.06, 95% CI [-0.35, 0.24], p = 0.705, Fig 5B); although the test for a disease status–by-methylation interaction was not significant (p = 0.12). No significant correlations were observed in cerebellum (S3 Fig).

Using the methylation levels at each of the 27 individual CpG sites of DMR 1, we observed negative correlations in the frontal lobe at 16 sites with unadjusted significant correlations (p<0.05) and another 5 sites exhibiting a similar trend (p-value between 0.05 and 0.07), but

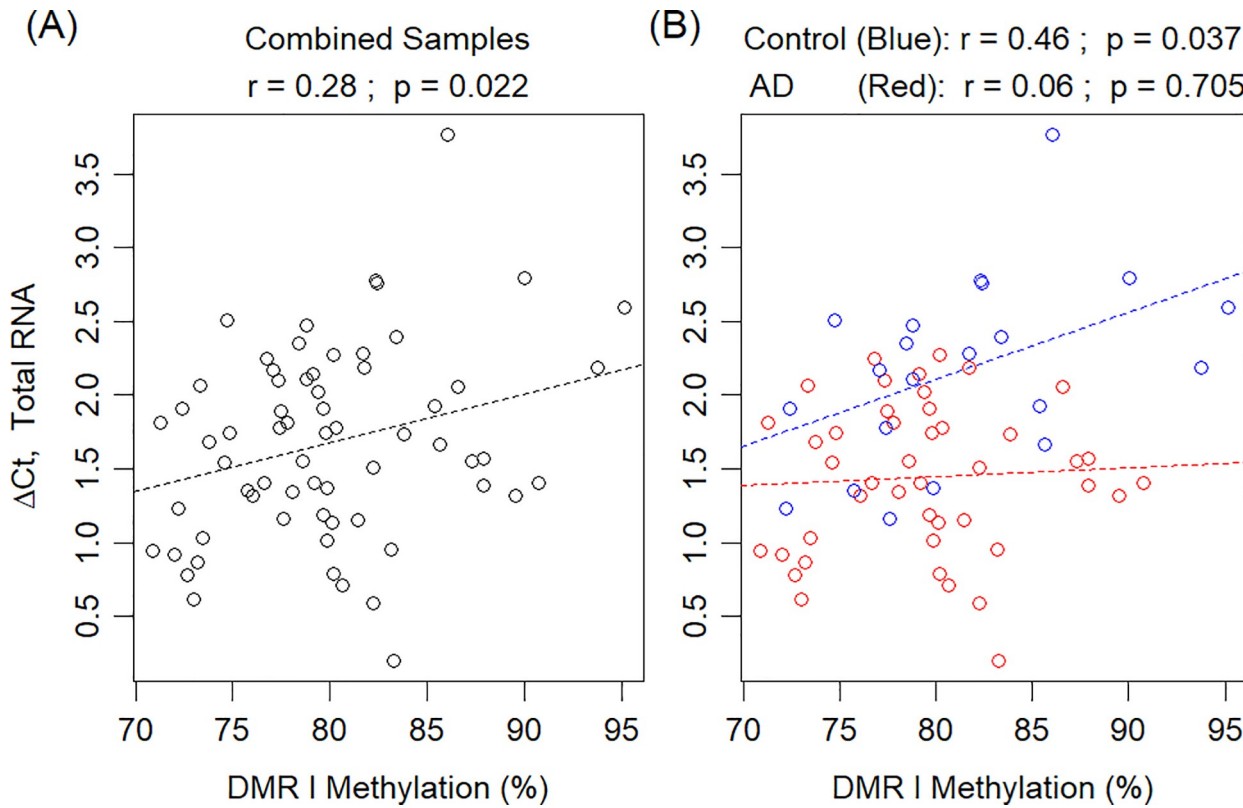

**Fig 5. Total RNA expression versus DNA methylation levels in frontal lobe.** Total *APOE* RNA ΔCt is plotted against mean methylation across DMR 1 (CpG site #11–37). (A) All frontal lobe samples (includes both AD and control subjects). (B) Plot separating AD (red) from control (blue). Dashed lines and p-values are associated with the respective fitted linear regression models. Note that lower ΔCt values represent higher expression levels. AD: Alzheimer's disease; Ct: cycle threshold; DMR 1: differentially methylated region 1.

none of these correlations were significant after adjusting for multiple comparisons. When separating AD and control subjects, unadjusted significant correlations are only present for control subjects. Examples of individual CpG site analyses in the frontal lobe are shown in S4 Fig. Again, no significant correlations were observed in cerebellum (S5 Fig). Together, these results imply that higher levels of DNA methylation at the CGI are correlated with lower expression of *APOE* RNA in the frontal lobe of control PMB. However, this relationship appears to be altered in the frontal lobe of AD PMB.

Methylated CGIs are known to have elevated binding of methyl-CpG binding proteins [56, 57]; thus, we speculated that protein binding at the *APOE* CGI could influence *APOE* transcription. To test this possibility, we first assessed the effect of altered DNA methylation levels at the *APOE* CGI on *APOE* RNA production. We applied 5-aza-dC treatment in human cell lines (HepG2, LN-229 [glioma cells], and SH-SY5Y). The chemical 5-aza-dC is a suicidal analog of cytosine, acting as a DNA methyltransferase (DNMT) inhibitor [58]. DNMT transfers a methyl group to C5 position of Cytosine at CpG dinucleotides. When 5-aza-dC is incorporated into DNA, a covalent bond is formed between 5-aza-dC and DNMT. Because C5 position of cytosine is replaced with nitrogen in 5-aza-dC, the methyl transfer reaction cannot take place and the DNMT is irreversibly bound to the DNA. The treatment of 5-aza-dC results in depleting pool of DNMT, leading to decrease in DNA methylation. Also, U87 produces low levels of *APOE* RNA, we have instead used another glioblostoma cell line LN-229 (with higher *APOE* RNA expression than U87) for this experiment. Following 5-aza-dC treatment, we quantified

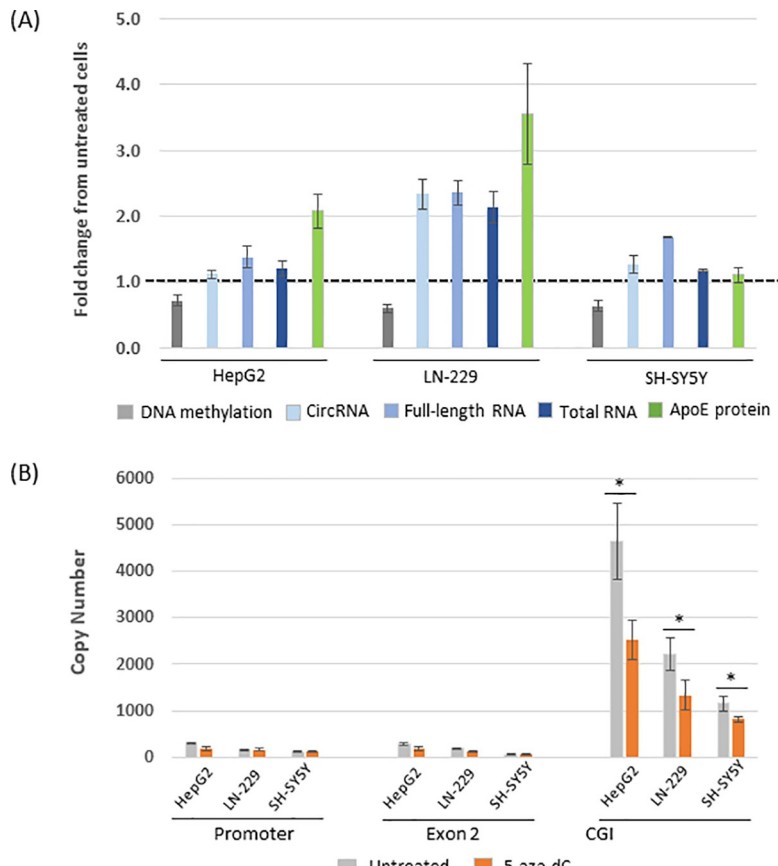

**Fig 6. Effects of DNA methylation, RNA/protein production, and protein binding in 5-aza-dC treated cell lines HepG2, LN-229, and SH-SY5Y.** (A) Comparison of *APOE* DNA methylation and *APOE* RNA expression and secreted ApoE levels in 5-aza-dC treated cells. Levels from treated cells are plotted as fold change when compared with their untreated counterparts (set at 1.0, dashed line). The DNA methylation levels at the *APOE* CGI were quantified using bisulfite pyrosequencing. The expression levels of *APOE* RNAs (circRNA, full-length RNA, and total RNA) were quantified by qRT-PCR using TaqMan assay. The expression levels of secreted ApoE proteins were quantified by ELISA. (B) Binding of MECP2 protein in three *APOE* gene regions (promoter, exon 2, and CGI) between 5-aza-dC treated and untreated cells using ChIP-qPCR. Error bars represent three independent assays. *: t-test, $p < 0.05$. CGI: CpG island; circRNA: circular RNA; qRT-PCR: quantitative reverse transcriptase polymerase chain reaction; ChIP: chromatin immunoprecipitation. ELISA: enzyme-linked immunosorbent assay.

DNA methylation levels at *APOE* DMR 1 using pyrosequencing, and quantified *APOE* RNA levels by quantitative RT-PCR (qRT-PCR) using TaqMan assay. In parallel, we quantified secreted levels of ApoE in cell culture medium using an ELISA. For these analyses, two independent cultures were used for bisulfite pyrosequencing, three independent qRT-PCR assays were performed, and media from three independent cultures (per cell-type and treatment combination) was assayed by ELISA. Average and standard deviation of these multiple assays are calculated and fold change of levels of 5-aza-dC treated cells compared to those of untreated cells (set at 1.0 for untreated) are used for plotting in Fig 6A.

We observed a marked reduction (28 to 40%) of methylation at DMR 1 for the treated cells compared to the untreated cells (Fig 6A). This decrease in DNA methylation was consistently associated with increased production of *APOE* RNAs (circRNA, full-length mRNA, and total RNA). Increase in *APOE* RNA production after 5-aza-dC treatment is confirmed with increased production of ApoE protein compared to the untreated cells. Very low expression of *APOE* gene in neuron cells is likely to account for almost no increased production of ApoE

protein from SH-SY5Y cells when compared to the untreated cells. Together, this result further supports the hypothesis that DNA methylation levels at the *APOE* CGI are associated with *APOE* RNA production.

To explore this association, we inspected the *APOE* CGI's DNA-protein interactions using chromatin immunoprecipitation quantitative PCR (ChIP-qPCR). We selected methyl CpG-binding protein 2 (MECP2) for this experiment because MECP2 can bind to methylated CpGs and negatively regulate gene expression [59, 60]. We performed ChIP on the three human cell lines (HepG2, LN-229, and SH-SY5Y) using an anti-MECP2 antibody, and we then PCR quantified ChIP-recovered DNA fragments of three *APOE* gene regions (the promoter, exon 2, and CGI). In untreated cells, the *APOE* CGI had an increased binding of MECP2 (~10 to 20-fold higher) compared to the *APOE* promoter and exon 2 (Fig 6B). In the 5-aza-dC treated cells, which had a significant reduction (28 to 40%) of DNA methylation at the *APOE* CGI, we observed reduced binding of MECP2 (~29 to 46%) at the *APOE* CGI compared to the untreated cells. Together, these results suggest that the hypermethylated *APOE* CGI can recruit methyl-CpG binding protein(s) to change the dynamics of *APOE* RNA transcription.

## Discussion

*APOE* is the strongest known genetic risk factor for late-onset AD. From a genetics perspective, a gene strongly associated with a disease traditionally plays a direct role in the pathogenesis of that disease. Such an effect is usually carried out by the gene's products including RNA and protein and can be explained by a change in either their qualities (structure or function) or quantity (expression levels) that leads to physiological changes. Differential expression of *APOE*'s RNA and protein have been observed in AD, but *APOE* RNA levels do not always correlate with ApoE protein levels, and it is unclear whether AD subjects have elevated or decreased *APOE* expression [31–38].The aim of this study was to revisit *APOE*'s transcriptional pathway, define the *APOE* CGI's role in this pathway, and determine how the relationship between *APOE* transcription and the *APOE* CGI correlates with the risk of AD.

We have quantified *APOE* RNA from neuropathologically confirmed AD and control human PMBs. Our data show that total *APOE* RNA has higher expression levels in frontal lobe from AD patients than from controls, which is consistent with the literature showing elevated *APOE* RNA in AD PMB [30, 35–38]. However, contrary to the commonly accepted paradigm that all *APOE* mRNA spans the full length of the gene, our study reveals that *APOE* RNA is composed of at least three distinct RNA species: circRNA, full-length mRNA, and truncated mRNA. We show that expression levels of the full-length mRNA constitute less than half of the total *APOE* RNA, with circRNAs and truncated mRNAs likely constituting most of the missing fraction. Given that only full-length mRNA is capable of being translated into the ApoE protein, including its lipid binding domain, this RNA should be considered the true protein-producing transcript of *APOE*. From this work, we can conclude that precise measurement of the full-length mRNA is crucial for studies aimed at correlating expression levels between *APOE*'s mRNA and protein.

Because it is not yet feasible to directly measure truncated *APOE* mRNAs, our observation that the levels of the *APOE* circRNA and full-length mRNA do not add up to the levels of total *APOE* RNA indirectly demonstrates the presence of truncated *APOE* mRNA, a view that is supported by the Ensembl RNA map [55] of *APOE*. These truncated mRNA transcripts do not appear to be in an unstable transient state, as they are robustly detected across PMB tissues. It is thus plausible that these RNAs could behave like noncoding RNA and play a role in regulating other genes/proteins or in providing a template for small peptide production. Further investigation into these transcripts is warranted.

The proportions of various *APOE* RNAs relative to *APOE* total RNA are similar across different brain tissues, suggesting that these RNAs are produced from a unified transcriptional pathway. The production of different mRNA isoforms from the same gene may arise through the usage of alternative transcriptional start sites, splicing sites, poly-A sites, or 3'UTRs. However, these well-defined mechanisms do not appear to be the cause of these prematurely terminated *APOE* mRNAs. Instead, the CGI present in the 3' ORF region in which the truncated *APOE* mRNA terminates may provide a plausible explanation. Along these lines, we found that lower levels of DNA methylation at the *APOE* CGI are associated with higher expression levels of *APOE* RNA in the control frontal lobe and in 5-aza-dC treated cells, suggesting that DNA methylation at the *APOE* CGI can influence *APOE* RNA production either directly or indirectly. Although our RNA-expression and DNA-methylation correlation analysis of AD frontal lobe did not exhibit this association, it is relevant to note that AD frontal lobe has lower DNA methylation levels at the *APOE* CGI compared to control frontal lobe [47] and that this AD tissue also exhibits increased *APOE* mRNA expression compared to control tissue. Together, these results suggest that additional disease-related factor may be modifying *APOE*'s transcriptional pathway in AD. Whether this AD-specific change is a cause, or a consequence of AD pathogenesis is unclear and warrants further investigation.

Additionally, in our ChIP experiments, hypermethylated *APOE* CGI showed an elevated binding of MECP2 protein that is associated with lower *APOE* RNA production. It is conceivable that this hypermethylated *APOE* CGI attracts binding of multiple methyl-CpG binding proteins [56], these proteins may then occupy the DNA template of the ORF, hindering the kinetics of the RNA Poly II complex and subsequently leading to either incomplete elongation of the mRNA (i.e., truncated mRNA) or generation of the circRNA. Together, these results reflect a clear relationship between methylation at the *APOE* CGI and *APOE* RNA production, a relationship that signifies epigenetically imparted mRNA regulation at the 3'-end of *APOE*.

In this study, we did not design comprehensive experiments to search for other non-coding RNA of *APOE*; however, there are evidences of their existence. A natural antisense *APOE* RNA transcript has been reported in the study of Seitz et al. [61], and a long non-coding *APOE* RNA (lnc-ZNF296-6) has been defined in a lncRNA database (LNCipdia). To the best of our knowledge, no known *APOE* microRNA has been identified. We queried multiple miRNA databases (miRbase, MiRDB, TarBase) and did not find any record of microRNA originating from the *APOE* gene.

Although the presence of *APOE* circRNAs has been previously proposed using transcriptome analyses [62, 63], we were surprised to discover the existence and abundance of these *APOE* circRNAs in human tissues. CircRNAs are part of a group of stable and evolutionary conservative noncoding RNAs, a group that has been recently identified as active in the post-transcriptional regulation of gene expression. Unlike linear RNAs, circRNAs lack a 5' cap and 3' poly(A) tail and have their 3' and 5' ends covalently linked together in closed circular loops by back splicing [64]. Back spliced exons using exon/intron junctions and base pairing between repeat elements are the most common mechanism for exonic circRNA formation [65–67]. However, *APOE* circRNAs do not seem to use the exon/intron junctions for back-splicing; thus, alternative mechanisms must be involved. *APOE* circRNA back-spliced sites share a common motif (AGCTGC) with the potential to form base pairings and stem-loop structures, which have been predicted by RNAstructure program [68] with high probability (S1 Fig). Because no known protein binding has been assigned to this motif, it remains to be resolved experimentally whether such stem-loop formation is the initial step and what proteins are involved in the circularization.

At present, the function of these *APOE* circRNAs is unknown, but their predominant presence in the cytoplasm suggests a post-transcriptional role. Both L- and S-circRNA contain

nucleotide numbers that can be divided by three, representing ORFs without a stop or start codon; they correspond, respectively, to mature ApoE protein amino acid 16–145 (L-circRNA) and 53–145 (S-circRNA). If these circRNA generate peptides, which would be translated through a rolling circle mechanism [35,36], these peptides could potentially modify the binding kinetics between *APOE* and the low-density lipoprotein receptor. Like the full-length *APOE* mRNA, *APOE* circRNAs are expressed at higher levels in AD frontal lobe compared to control frontal lobe, and when comparing the difference in expression levels between AD and control subjects, *APOE* circRNAs and total *APOE* RNA have a similar effect size. These results suggest that *APOE* circRNAs may indeed play a role in AD risk. It is also interesting to note that they carry the ε4 versus ε2/ε3 variants, implying the potential for ε4-linked effects in AD. Whether these *APOE* circRNAs have an independent biological effect in AD warrants further investigation. Compared to mRNA and other linear noncoding RNA, circRNA have higher stability and a longer half-life [64], and their small nature enables them to serve as quick-response molecules for epigenetic cues (e.g., changes in environment or lifestyle). Therefore, *APOE* circRNAs could potentially be developed as a new biomarker for AD to monitor disease progression and/or intervention.

## Conclusions

Our findings define a new paradigm of *APOE* gene regulation and provide novel insight into the transcriptional pathway of *APOE*. This pathway involves not only the production of multiple *APOE* RNA species but also an epigenetically imparted transcriptional program driven by the *APOE* CGI. These findings can be built upon to stimulate the discovery of new drug targets and the development of more precise *APOE*-based intervention and/or therapeutic strategies for AD. This knowledge could also inform studies of other *APOE*-linked disorders.

## Supporting information

**S1 Fig. Sequence and predicted stem-loop formation of *APOE* circRNA.** (A) Partial sequence of *APOE* Ex 3 (blue font) and 4 (black font) shows locations of regular splice site, back-splice sites that contain a common AGCTGC sequence (highlighted in yellow), and the two outward primers (orange arrows) that amplify *APOE* circRNAs. Note that both circRNAs contain the ε4-determing SNP (rs429358, red font with underline). (B) Stem-loop formation of L-circRNA from original template was predicted using the RNAstructure program (https://rna.urmc.rochester.edu/RNAstructureWeb) with a probability of >75%. (C) The same analysis predicted a probability of >95% for the stem-loop formation of S-circRNA. circRNA: circular RNA; Ex: exon; L: large; S: small; SNP: single-nucleotide polymorphism.
(PDF)

**S2 Fig. Correlations of expression levels of *APOE* RNA types.** Expression levels of three *APOE* RNAs (circular, full-length, and total) are plotted as values of ΔCt (Ct of *APOE* RNA–Ct of *ACTB* RNA) in both frontal lobe (left panel) and cerebellum (right panel). Plots on the diagonal are the empirical density of expression levels for that particular RNA type. Corr: correlation; Ct: cycle threshold.
(PDF)

**S3 Fig. Total RNA expression versus DNA methylation levels in cerebellum.** Total *APOE* RNA ΔCt is plotted against mean methylation across DMR I (CpG site #11–37). (A) All cerebellum samples including both AD and control subjects. (B) Plot separating AD (red) from control (blue). Dashed lines and p-values are associated with the respective fitted linear regression models. Note that lower ΔCt values represent higher expression levels. AD: Alzheimer's

disease; Ct: cycle threshold; DMR I: differentially methylated region 1.
(PDF)

**S4 Fig. Total RNA expression versus DNA methylation levels in frontal lobe.** Total *APOE* RNA ΔCt is plotted against DNA methylation levels of individual *APOE* CpG sites (#19, 21, and 29) for all frontal lobe samples (includes both AD and control samples; left panel) and separated AD (red) and control (blue) samples (right panel) with respective linear fit lines (dashed) and uncorrected correlation p-values. Note that lower ΔCt values represent higher expression levels. AD: Alzheimer's disease; Ct: cycle threshold; Ctrl: control.
(PDF)

**S5 Fig. Total RNA expression versus DNA methylation levels in cerebellum.** Total *APOE* RNA ΔCt is plotted against DNA methylation levels of individual *APOE* CpG sites (#19, 21, and 29) for all cerebellum samples (includes both AD and control samples; left panel) and separated AD (red) and control (blue) samples (right panel) with respective linear fit lines (dashed) and uncorrected correlation p-values. Note that lower ΔCt values represent higher expression levels. AD: Alzheimer's disease; Ct: cycle threshold; Ctrl: control.
(PDF)

**S1 Table. Primers, probes, and TaqMan assays.**
(PDF)

**S2 Table. Fraction of *APOE* RNA types in PMB tissue.**
(PDF)

## Author Contributions

**Conceptualization:** Eun-Gyung Lee, Jessica Tulloch, Chang-En Yu.

**Data curation:** Eun-Gyung Lee, Jessica Tulloch, Chang-En Yu.

**Formal analysis:** Kaitlin Todd, Steve Millard.

**Funding acquisition:** Chang-En Yu.

**Investigation:** Eun-Gyung Lee, Jessica Tulloch, Sunny Chen, Lesley Leong, Aleen D. Saxton.

**Methodology:** Eun-Gyung Lee, Sunny Chen, Lesley Leong, Chang-En Yu.

**Project administration:** Chang-En Yu.

**Resources:** Brian Kraemer, Martin Darvas, C. Dirk Keene.

**Supervision:** Chang-En Yu.

**Validation:** Eun-Gyung Lee, Chang-En Yu.

**Visualization:** Eun-Gyung Lee, Jessica Tulloch, Andrew Shutes-David, Chang-En Yu.

**Writing – original draft:** Chang-En Yu.

**Writing – review & editing:** Eun-Gyung Lee, Jessica Tulloch, Andrew Shutes-David, Kaitlin Todd, Steve Millard, Chang-En Yu.

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
