## [Decision Letter · Decision Letter 0]

7 Nov 2019

PONE-D-19-25663

Redefining transcriptional regulation of the APOE gene and its association with Alzheimer’s disease

PLOS ONE

Dear Dr. Yu,

Thank you for submitting your manuscript to PLOS ONE. After careful consideration, we feel that it has merit but does not fully meet PLOS ONE’s publication criteria as it currently stands. Therefore, we invite you to submit a revised version of the manuscript that addresses the points raised during the review process.

We would appreciate receiving your revised manuscript by Dec 22 2019 11:59PM. To enhance the reproducibility of your results, we recommend that if applicable you deposit your laboratory protocols in protocols.io, where a protocol can be assigned its own identifier (DOI) such that it can be cited independently in the future. For instructions see: http://journals.plos.org/plosone/s/submission-guidelines#loc-laboratory-protocols

We look forward to receiving your revised manuscript.

Kind regards,

Y-h. Taguchi, Dr. Sci.

Academic Editor

PLOS ONE

Journal Requirements:

2. Our internal editors have looked over your manuscript and determined that it may be within the scope of our Early Diagnosis and Treatment of Alzheimer's Disease Call for Papers. This collection of papers is headed by a team of Guest Editors for PLOS ONE: Michael Weiner, Roberta Brinton, Jussi Tohka and Yona Levites. With this Collection we hope to bring together researchers working on a wide range of disciplines, from molecular and preclinical work, through to patient-centered studies, including clinical trials.   Additional information can be found on our announcement page: https://collections.plos.org/s/alzheimersdisease. If you would like your manuscript to be considered for this collection, please let us know in your cover letter and we will ensure that your paper is treated as if you were responding to this call. Agreeing to be part of the call-for-papers will not affect the date your manuscript is published. If you would prefer to remove your manuscript from collection consideration, please specify this in the cover letter.

3. We noticed you have some minor occurrence of overlapping text with the following previous work, which needs to be addressed:

https://www.degruyter.com/view/j/bmc.2015.6.issue-1/bmc-2014-0039/bmc-2014-0039.xml

https://academic.oup.com/hmg/article/22/24/5036/569347

In your revision ensure you cite all your sources (including your own works), and quote or rephrase any duplicated text outside the methods section. Further consideration is dependent on these concerns being addressed.

Additional Editor Comments:

Two reviewers evaluated the manuscript positively. All the concerns raised are not critical. Please submit revised version with considering all the concerns raised.

Reviewers' comments:

Reviewer's Responses to Questions

**Comments to the Author**

1. Is the manuscript technically sound, and do the data support the conclusions?

Reviewer #1: Yes

Reviewer #2: Partly

2. Has the statistical analysis been performed appropriately and rigorously? 

Reviewer #1: Yes

Reviewer #2: I Don't Know

3. Have the authors made all data underlying the findings in their manuscript fully available?

Reviewer #1: Yes

Reviewer #2: Yes

4. Is the manuscript presented in an intelligible fashion and written in standard English?

Reviewer #1: Yes

Reviewer #2: Yes

5. Review Comments to the Author

Reviewer #1: Around 40 million people over the age of 60 are suffering from Alzheimer's disease (AD) all over the world, but the mechanism and therapy of AD remain elusive. Apolipoprotein E (APOE) is the strongest genetic risk factor for AD. Chang-En Yu’s group presented an article entitled “Redefining transcriptional regulation of the APOE gene and its association with Alzheimer’s disease”. In this article, the authors identified novel APOE RNA transcripts, quantified their expression in human postmortem brain (PMB), and discovered a correlation between RNA expression and DNA methylation. This article is well-organized, but there are still some questions need to be addressed further.

1. It was very interesting that the authors identified two novel circRNAs (termed as S-circRNA and L-circRNA) using a genomic walking strategy in conjunction with RT-PCR. As we all known, there are other non-coding RNAs besides circRNAs, like microRNAs and lncRNAs. So, are there any reports about the microRNAs or lncRNAs of APOE? Or have the authors ever done any related tests of microRNAs or lncRNAs of APOE although the authors shortly addressed the question in discussion?

2. The authors used the human cell lines, including HepG2 (hepatocellular carcinoma cells), SH-SY5Y (neuroblastoma cells), and U87 (glioma 164 cells) to quantified APOE RNAs. The rationale to choose these three cell lines should be clarified more directly in text.

3. In line 171, “we used antibodies specific to either the N-terminus or the C-terminus of the APOE protein and performed semiquantitative Western blot analysis on lysates from HepG2 cells to inspect whether the truncated mRNAs can be translated into proteins.” The results in SH-SY5Y and U87 cell lines should be added in the article.

4. In line 304, the usage of 5-Aza-dCTP for the inhibition of DNA methylation should be addressed clearly.

5. In lines 308-310 and figure 6, the western blot of ApoE protein should be also used to demonstrate the effects of 5-Aza-dCTP treatment.

6. In line 317, “We performed ChIP on HepG2 cells using an anti-MECP2 antibody, and we then PCR quantified ChIP-recovered DNA fragments of three APOE gene regions.” The results in SH-SY5Y and U87 cell lines should be also added and discussed in the article.

7. Women have higher risk to develop AD than men. Epidemiological studies revealed that two-thirds of AD patients are women. In line 259, “When APOE ε4 status and sex were included as covariates, effect sizes increased slightly for all three RNAs, though the increases were more pronounced for circRNA (from 1.21 to 1.45) and 260 total RNA (from 1.32 to 1.43) than for full-length mRNA (from 1.04 to 1.05).” Can the authors make some explanations about the correlation between the gender difference and APOE RNAs expression level differences?

8. The authors selected frontal lobe and cerebellum for this experiment because the frontal lobe is heavily affected by AD pathology whereas the cerebellum is minimally affected. As we all known, hippocampus plays a key role in learning and memory, which defected in AD patients. So, have the authors ever detected the expression of APOE RNAs in hippocampus? Or are there any reports in hippocampus?

9. Most of the references were outdated. So updated references should be cited For example, “Grimm, A., Mensah-Nyagan, A. G. & Eckert, A. Alzheimer, mitochondria and gender. Neurosci. Biobehav. Rev. 67, 89–101 (2016)”.

10. Lack of references in several parts (for example, line 326-331).

Reviewer #2: In this study, Lee et al. characterized different types of APOE transcripts through analysis of cell lines and frozen frontal cortex of AD subjects & controls. This was further complemented with APOE methylation analysis and as well as the identification of APOE circRNAs that may contribute to transcriptional regulation of APOE. The assessments are thorough and findings from this study are important and valuable to share with the community. Our understanding of how APOE contributes to AD pathogenesis remains wanting and this study’s findings will help to address this gap. I have comments below to further clarify the manuscript:

Major concerns:

1) The authors state in the Author summary, Intro & Discussion that they assess how the association between APOE transcription and methylation influences the risk of AD, but this specific analysis is not performed and should be reworded.

2) Results (lines 169-175): the authors conclude that since they did not observe truncated ApoE protein, that only full-length APOE mRNA can be translated into functional ApoE protein. This conclusion is only derived from the analysis of HepG2 cells – was this analysis similarly performed in SH-SY5Y and U87 cells and/or human samples since it is possible that truncated ApoE protein may be expressed in other cell lines or sample types? (and especially since the full-length mRNA constituted lower proportions of total RNA for SH-SY5Y & U87 cells)

3) Results (lines 169-175): for the Western blot assessments, does the C-terminus antibody target all 3 isoforms?

4) Results (lines 190-191): was there evidence of APOE circRNAs in all PMB samples since only some are shown in Fig3?

5) Results (line 198): if the e4-determining SNP is present in both circRNAs, that seems to suggest that only e4 transcripts could be transcriptionally regulated by the circRNAs. Is this correct? This would be relevant for the results described in lines 258-262.

6) Results (lines 301-311): was this analysis performed on any PMB samples?

7) Discussion (lines 328-329): pathogenic effects are not just associated with quality or quantity, but more importantly, functional changes, which may not necessarily be associated quality or quantity.

8) Please provide methods text on the methylation analyses, including the sites assessed for the DMR 1 analyses and the genomic walking analyses. Please also clarify the number of replicates (biological & technical) used for each analysis.

Minor concerns:

1) Something to consider is that the cellular content of analyzed samples is a key factor as well for the methylation and expression analyses since APOE is primarily expressed in astrocytes and microglia.

2) Fig1: what transcripts are associated with the e2, e3, and e4 isoforms?

3) As there are public RNAseq data on AD brains, it would be interesting to see if the identified circRNAs are present in those datasets.

4) Please provide statistics behind the analysis of reduced methylation & increased mRNA production (lines 301-311).

5) There are a few grammatical errors throughout.

6. PLOS authors have the option to publish the peer review history of their article (what does this mean?). If published, this will include your full peer review and any attached files.

Reviewer #1: No

Reviewer #2: No

---

## [Author Response · Author response to Decision Letter 0]

13 Dec 2019

We have submitted a "Author response to reviewers' comments" letter with a point by point discussion of Reviewers’ comments.

---

## [Decision Letter · Decision Letter 1]

27 Dec 2019

Redefining transcriptional regulation of the APOE gene and its association with Alzheimer’s disease

PONE-D-19-25663R1

Dear Dr. Yu,

We are pleased to inform you that your manuscript has been judged scientifically suitable for publication and will be formally accepted for publication once it complies with all outstanding technical requirements.

With kind regards,

Y-h. Taguchi, Dr. Sci.

Academic Editor

PLOS ONE

Additional Editor Comments (optional):

This paper was accepted in the present form, but some errors pointed by reviewers should be corrected prior to production. Congratulations!

Reviewers' comments:

Reviewer's Responses to Questions

**Comments to the Author**

1. If the authors have adequately addressed your comments raised in a previous round of review and you feel that this manuscript is now acceptable for publication, you may indicate that here to bypass the “Comments to the Author” section, enter your conflict of interest statement in the “Confidential to Editor” section, and submit your "Accept" recommendation.

Reviewer #1: All comments have been addressed

Reviewer #2: All comments have been addressed

2. Is the manuscript technically sound, and do the data support the conclusions?

Reviewer #1: Yes

Reviewer #2: Yes

3. Has the statistical analysis been performed appropriately and rigorously? 

Reviewer #1: Yes

Reviewer #2: Yes

4. Have the authors made all data underlying the findings in their manuscript fully available?

Reviewer #1: Yes

Reviewer #2: Yes

5. Is the manuscript presented in an intelligible fashion and written in standard English?

Reviewer #1: No

Reviewer #2: Yes

6. Review Comments to the Author

Reviewer #1: The authors addressed the comments very well. However, there are still a couple of minor comments:

1. Please use standard gene and protein nomenclature: human genes (uppercase and italic), human proteins (uppercase), mouse genes (first letter only uppercase and italic) and mouse proteins (uppercase).

2. Line 458: Cytosine should be cytosine.

3. Please double check the manuscript to meet the typographic style.

Reviewer #2: There are distributed grammatical errors in the edits that were made in the revision. These should be corrected for clarity but the authors have addressed all of my other concerns.

7. PLOS authors have the option to publish the peer review history of their article (what does this mean?). If published, this will include your full peer review and any attached files.

Reviewer #1: No

Reviewer #2: No

---

## [Editor Report · Acceptance letter]

17 Jan 2020

PONE-D-19-25663R1 

Redefining transcriptional regulation of the APOE gene and its association with Alzheimer’s disease 

Dear Dr. Yu:

I am pleased to inform you that your manuscript has been deemed suitable for publication in PLOS ONE. Congratulations! Your manuscript is now with our production department. 

With kind regards,

on behalf of

Professor Y-h. Taguchi 

Academic Editor

PLOS ONE